# Use of a DNN in Recording and Analysis of Operator Attention in Advanced HMI Systems

Zbigniew Gomolka [1],*[iD], Ewa Zeslawska [1],*[iD], Boguslaw Twarog [1][iD], Damian Kordos [2][iD] and Pawel Rzucidlo [2][iD]

1   College of Natural Sciences, University of Rzeszow, Pigonia St. 1, 35-959 Rzeszow, Poland
2   Department of Avionics and Control, Faculty of Mechanical Engineering and Aeronautics, Rzeszow University of Technology, al. Powstancow Warszawy 12, 35-959 Rzeszow, Poland
*   Correspondence: zgomolka@ur.edu.pl (Z.G.); ezeslawska@ur.edu.pl (E.Z.)

**Abstract:** The main objective of this research was to propose a smart technology to record and analyse the attention of operators of transportation devices where human–machine interaction occurs. Four simulators were used in this study: General Aviation (GA), Remotely Piloted Aircraft System (RPAS), AS 1600, and Czajka, in which a spatio-temporal trajectory of system operator attention describing the histogram distribution of cockpit instrument observations was sought. Detection of the position of individual instruments in the video stream recorded by the eyetracker was accomplished using a pre-trained Fast R-CNN deep neural network. The training set for the network was constructed using a modified Kanade–Lucas–Tomasi (KLT) algorithm, which was applied to optimise the labelling of the cockpit instruments of each simulator. A deep neural network allows for sustained instrument tracking in situations where classical algorithms stop their work due to introduced noise. A mechanism for the flexible selection of Area Of Interest (AOI) objects that can be tracked in the recorded video stream was used to analyse the recorded attention using a mobile eyetracker. The obtained data allow for further analysis of key skills in the education of operators of such systems. The use of deep neural networks as a detector for selected instrument types has made it possible to universalise the use of this technology for observer attention analysis when applied to a different objects-sets of monitoring and control instruments.

**Keywords:** eye tracking; deep neural network; attention trajectory; HMI systems

## 1. Introduction

The eye-tracking equipment market lacks objective systems to determine the level of training of those operating multitasking mechanical equipment (and more). Existing mobile eyetracker software does not provide seamless AOI analysis in recorded video streams. The products of leading eye-tracking system manufacturers do not provide such functionality [1–3]. This paper extends earlier research's related with pilot attention tracking during key procedures, i.e., take-off and landing, which was carried out using the SMI RED 500 and Tobii T60 stationary eyetrackers and the Tobii Glasses mobile eyetracker. Stationary eyetrackers allow video streams to be recorded and further analysed using the environments provided by SMI—BeGaze and Tobii Studio. In addition, the analysed example of the study of the assessment of the relationship between eye-tracking measurements and the perceived workload in robotic surgical tasks was the problem described in the article [4]. Another analysed study aimed to review eye-tracking concepts, methods and techniques by developing efficient and effective modern approaches such as machine learning (ML), Internet of Things (IoT) and cloud computing. These approaches have been in use for over two decades and are used heavily in the development of the latest eye-tracking applications [5]. In another study, the authors developed three artificial intelligence techniques, namely machine learning, deep learning, and a hybrid technique between them, for the early diagnosis of autism. The first technique, feedforward neural networks (FFNN) and

artificial neural networks (ANN), the second technique using the pre-trained convolutional neural network model (CNN, GoogleNet and ResNet-18), and the third technique used the hybrid method between deep learning (GoogleNet and ResNet-18) and machine learning (SVM), called GoogleNet + SVM and ResNet-18 + SVM, and these achieved high performance and accuracy [6]. For example, work to develop observer attentional statistics for a 3 min video sequence (MJPG2000, 640 × 480 pixels at 30 fps) covering the location of nine instruments in the cockpit requires at least 48 h of user effort see Figure 1.

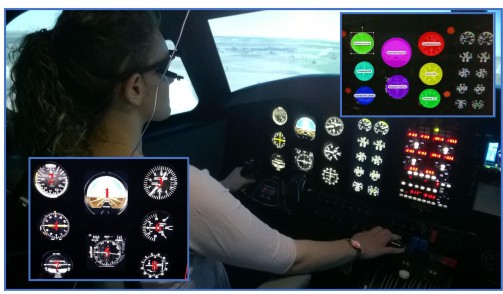

**Figure 1.** General view of the pilot attention measurement strategy during fly.

The effect of this engagement is one-off and requires redefining the AOI both when changing the operator and the flight task being recorded. This functionality is important, especially in the case of complex Human–Machine Interfaces (HMI) systems, where it is indispensable to transfer large amounts of data in the shortest possible time, e.g., production line operators or operators of ground air traffic control stations of unmanned systems [7–11]. The approach proposed in this paper is based on well-known and used eye-tracking systems which, due to their universality and targeting a different audience, do not allow for application in the areas and tasks that are the purpose of this paper. The authors hypothesise that the use of a deep pre-trained neural network supporting a mobile eye-tracking system will enable both the intelligent location of AOI areas throughout the video sequence and the obtaining of appropriate observation histograms for individual AOIs defined for a given HMI.

The process of detecting the position of individual instruments in the recorded video stream was carried out using the pre-trained Fast R-CNN deep neural network. The training vector for the network was realised using a modified KLT algorithm, which optimised the labelling of cockpit instruments. A deep neural network allows you to keep track of instruments in situations when classical algorithms stop their work due to digital noise. Among the available machine learning methods, we can also use the AutoML automated learning system. AutoML was designed as an artificial intelligence-based solution to the growing demand of applying machine learning. The high degree of automation in AutoML aims to allow non-experts to make use of machine learning models and techniques without requiring them to become experts in machine learning. As big data become ubiquitous across domains, and more and more stakeholders aspire to make the most of their data, demand for machine learning tools has spurred researchers to explore the possibilities of automated machine learning (AutoML). AutoML tools aim to make machine learning accessible for non-machine learning experts (domain experts), to improve the efficiency of machine learning, and to accelerate machine learning research. In this review article, the authors introduce a new classification system for AutoML systems, using a seven-tiered schematic to distinguish these systems based on their level of autonomy [12]. In another article, the authors examine the readiness of popular AutoML frameworks from the perspective of machine learning practitioners. Their goal is to demonstrate how the growing AutoML trend will affect the future job responsibilities of scientists, researchers, and human data practitioners [13]. In article [14], the authors try to investigate the interaction between data cleansing and other ML pipeline hyperparameters for supervised binary classification tasks. They use AutoML for both the dirty and clean state of all CleanML datasets. Because AutoML optimises the entire pipeline, you can avoid measurement artifacts related to static

preprocessing. The method proposed in this paper for recording and analysing operator attention is a key added value and its light motive.

## 2. Research Methods

### 2.1. HMI Stations

For purposes of the research work carried out, workstations were selected and configured using HMIs in which it was possible to record the observer's attention. The data collected in this way will allow received video streams to be processed using a Deep Neural Network (DNN) and thus enable the level of training of the operators of selected HMI systems to be assessed [15–17]. In particular, this will apply to operators of manned and unmanned aircraft and operators of means of a road transport. For this purpose, four sets of devices using HMIs were prepared (see Figure 2):

- GA flight simulator equipped with standard analogue instrumentation, configured to type FNTP II MCC as a twin-engine piston aircraft, with analogue-equipped retractable landing gear, providing an alternative for pilots practically training in the air;
- Flying laboratory Czajka MP02A, which is a two-crew (pilot + passenger) high-wing monoplane with tricycle landing gear of carbon–polymer laminate construction (span 9.72 (m), width 1.215 (m)) with engine propulsion type Rotax 912 ULS with a traction propeller. An aeroplane adapted to and capable of providing flight tests of the avionics equipment and solutions was used. A laboratory equipped with a research control and navigation system was developed under the LOT project;
- RPAS ground station as a simulated electric-powered aircraft with a span of 2.6 (m) and take-off weight of 2.5 (kg);
- The AS 1600 truck simulator with a 6-degree-of-freedom motion platform (based on a SCANIA truck cab).

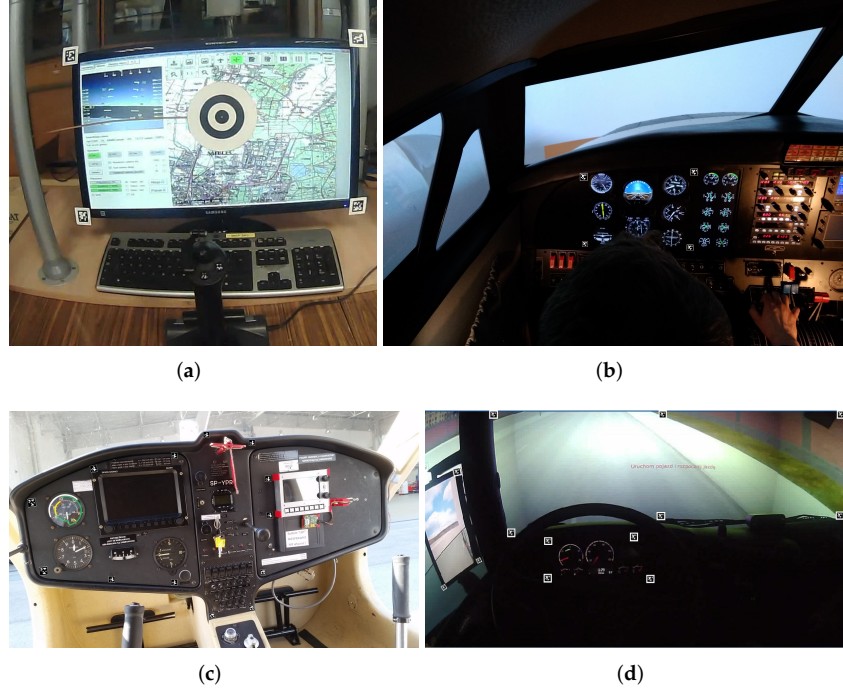

**Figure 2.** Observer attention recording and analysis stand (**a**) RPAS, (**b**) GA simulator (cockpit view), (**c**) flying lab (cockpit view), (**d**) truck simulator (cockpit view).

In the course of the research carried out on individual simulators, each station was equipped with a prototype of a mobile eye-tracking system along with dedicated components allowing for the process of calibration, validation and recording of video sequences.

### 2.2. An Eye-Tracking System for Attention Recording

The study used the open source eye-tracking system Pupil Core v2.0.182 equipped with a camera recording the observed scene and two cameras recording eye movement in infrared. This is a typical system that is subject to calibration at the initial stage and is a key element in determining the accuracy of the recorded coordinates of the operator's observations [18–21]. In order to achieve high accuracy of the attentional measurements, 5-point calibration using a screen, calibration using a calibration tag/marker and natural calibration made possible by the use of Apriltags were used interchangeably. Due to the nature of the research, calibration for the flight simulator and the Czajka aircraft was carried out using a calibration marker. The eye-tracking system employed the 2D Gaze Mapping eye coordinate detection model, which maintains accuracy within a visual error limit of <1°. The use of the 2D Gaze Mapping model works mainly in systems where participants do not have to move their heads and the experiment time is relatively short [22–25].

### 2.3. Tasks for HMI Station Operators

During the execution of each test, subjects in the experiments were asked to perform the following tasks:

- GA simulator. Execution of a precision instrument approach as indicated by the Instrument Landing System (ILS). This approach provides vertical and horizontal guidance to the Decision Altitude (DA). The barometric height DA is related to the local pressure prevailing at medium sea level—QNH (Q Nautical Height). Glide performed under minimum weather conditions for CAT I ILS, i.e., visibility along runway RVR 550 m and cloud base at 200 ft AGL (Above Ground Level). Atmospheric conditions allow flights only under IFR (Instrument Flight Rules), which means that pilots can navigate solely on the basis of on-board instruments. The flight takes place in windless conditions on a configured aircraft with three-point retractable landing gear. The aircraft configuration includes flaps in the landing position, landing gear released, and power close to minimum. Flights are performed at the EPRZ—Rzeszow Jasionka airport on runway 27 with an available landing length of 3192 m and width of 45 m. The airport elevation is 693 ft, and the magnetic heading of the runway is 265 degrees. The flight starts at 7 NM from the runway threshold and at a barometric altitude of 3000 ft ft AMSL (Above Medium Sea Level). The task ends when a Decision Height (DH) of 200 ft is reached.
- Flying laboratory Czajka MP02A. Flight operations under Visual Flight Rules (VFR). Flight operations were divided into three main stages: take-off from the EPRJ OKL airport (Aviation Training Centre) (magnetic runway heading 265 degrees), building a traffic pattern according to instructions from the air traffic control tower after reaching 1000 ft AMSL, and landing at the EPRJ airport. The aircraft on which the simulator tasks are performed is fully configured and adapted to perform a given phase of flight.
- RPAS ground station. Performing a flight in BVLOS mode (Beyond Visual Line of Sight—operations beyond the visual range of the UAV operator) in the EPML airport area. The system operator uses a hand-held mini control panel, an on-board camera image, an integrated pilot-navigation display system and an interactive map to perform a manual take-off. The take-off and initial climb was on the 09 direction. After reaching an altitude of 100 m AGL, a 90-degree turn to the left was followed by a course to the north and a further climb to an altitude of 150 (m) AGL. Then, there was another turn to the left on a course of 270 degrees in order to build a traffic pattern. This part of the flight was already in manual-assisted mode, which was characterised by the protection of the flight parameter envelope, in particular the angles of spatial orientation. After the third turn, the descent begins, which is followed by a final turn to bring the aircraft straight ahead for landing. The entire landing process, including the landing roll, was carried out in assisted manual mode.
- Truck simulator. Driving a lorry in urban conditions where during the journey, the driver pays attention to the correct observation of junctions and crossings with traffic

lights, pedestrians and other road users, and observes speed limits and road signs. A simulation was carried out under different weather and day conditions.

Once the group of people taking part in the experiment had been formed, the research was carried out on a test group of pilots and drivers with varying degrees of training and experience in aviation and operators of road transport vehicles. The experiment lasted no longer than 30 min in order to preserve the correct perception of the subject and to limit the influence of fatigue. This allowed them to focus their attention better. The research was conducted on two groups of people: in the flying task, Group I included people with less than 80 h of flying experience considered as a group of inexperienced pilots (NONPILOT). Group II (PILOT), on the other hand, is made up of people with >80 h flight experience. In the truck control task, a distinction was made between Group I (those with less than 10 h driving experience) and Group II (those with more than 10 h driving experience).

## 3. Operator Attention Tracking System Using a DNN

Figure 3 presents the modular structure of a designed workstation for recording and analysing the attention of operators of advanced systems using an HMI. Within Module A are sets of infrared sensors to detect the position and orientation of the observer's pupil (1) and scene observation cameras (2). The module is responsible for acquiring the video stream of the observed scene and recording the spatial position of the observer's pupils. Module B performs the tasks of attentional coordinate detection (3) and fixation detection (5). Module C performs AOI detection tasks in the video stream with the trained DNN [26–30]. Its main component (4) performs the task of detecting in the video stream the presence of instruments that are part of the HMI in use. This is completed by using the trained weight set obtained from the set of interface components (4a). The direct instrument recognition block (4b) in the video stream uses a deep neural network (4bb).

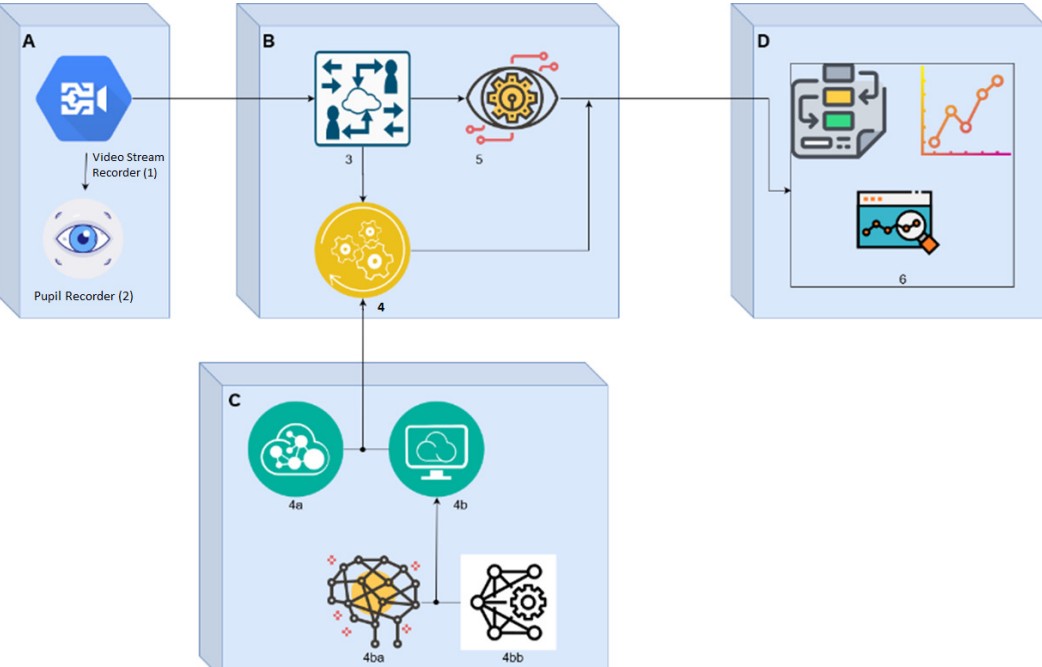

**Figure 3.** Schematic diagram of a modular workstation structure for recording and analysing operator attention in advanced HMI systems. (**A**–**D**) denotes consecutive modules for signal processing in the system.

The authors assumed that a set of pre-trained architectures provided by the DNN Toolbox Matlab® library environment would be used in the DNN simulation studies. A modified KLT algorithm (see Algorithm 1) was implemented to prepare a training set that would contain a series of video frames of the operator cockpit instruments of a given

station. Its task is to match the template image $T(x)$ with the input image $I(x)$. The vector $x$ contains the coordinates of the image $[x,y]T$.

---

**Algorithm 1** Image characteristic point detector tracker–KLT algorithm

---

Input: $I(x), \varepsilon$
Output: $p$
begin
do{

warp $I(x)$ with $W(x,p) \rightarrow I(W(x,p))$
the error $T(x) \rightarrow I(W(x,p))$
warped gradients $\nabla I = [I_x, I_y]$ evaluated at $W(x,p)$
the Jacobian of the warping $\frac{\partial W}{\partial p}$
steepest descent $\nabla I \frac{\partial W}{\partial p}$
inverse Hessian $H^{-1} = \left[ \sum_{x \in T} \left[ \nabla I \frac{\partial W}{\partial p} \right]^T \left[ \nabla I \frac{\partial W}{\partial p} \right] \right]^{-1}$
multiply steepest descend with error:
$\sum_{x \in T} \left[ \nabla I \frac{\partial W}{\partial p} \right]^T [T(x) - I(W(x,p))]$
compute $\Delta p$
update parameters $p \leftarrow p + \Delta p$

}**while** $(\Delta p < \varepsilon)$
end;

---

In matrix notation form, it can be represented accordingly:

$$W(x,p) = \begin{bmatrix} x + p_1 \\ x + p_2 \end{bmatrix} \tag{1}$$

$$\Delta p = H^{-1} \sum_{x \in T} \left[ \nabla I \frac{\partial W}{\partial p} \right]^T [T(x) - I(W(x,p))] \tag{2}$$

$$H = \sum_{x \in T} \left[ \nabla I \frac{\partial W}{\partial p} \right]^T \left[ \nabla I \frac{\partial W}{\partial p} \right] \tag{3}$$

In this form, the algorithm was introduced as a tracker function into VideoLabeler, which allowed the generation of a sequence of images into the GroundTruth variable that constitutes the training set for each simulator. It should be noted that if tracked points are lost from the field of view (see Figure 4), it is unable to re-determine their location when they reappear—hence the need to use machine learning for object recognition.

An example of video recording showing the loss of ROI3 when the operator performs hand movements—the lost Region of Interest (ROI) is marked with a dashed line (left bottom corner of the cockpit image on Figure 4). The data obtained in the marking session exported to the GroundTruth object (see Table 1, Figure 5) contain, among other things, information about the time (in seconds since the beginning of the video) and the position of rectangular object envelopes in the form of centre coordinates and dimensions (in pixels): $[x, y, W, H]$

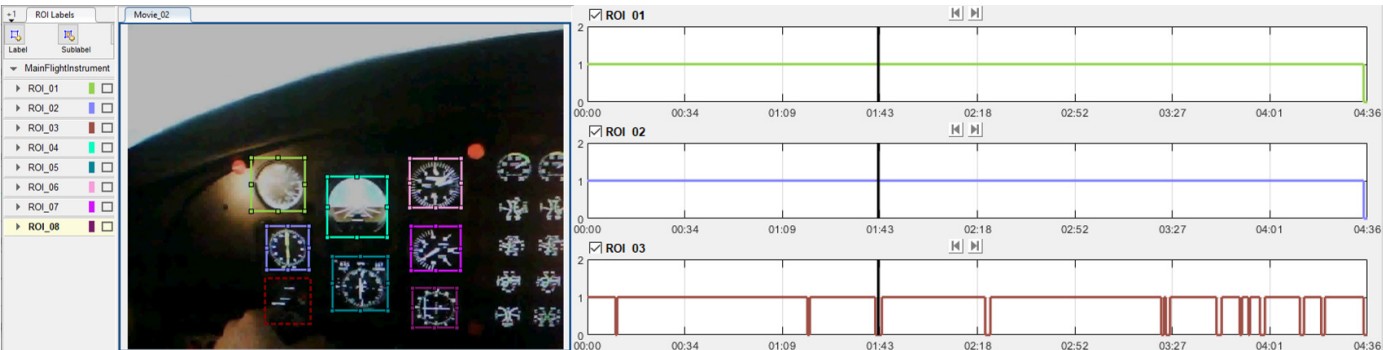

**Figure 4.** Example video recording showing the loss of ROI3 when the operator performs hand movements—the lost ROI is marked with a dashed line.

**Table 1.** GroundTruth with properties.

| Simulator Type | Data Source | Label Definitions | Label Data |
|---|---|---|---|
| RPAS | [1 × 1 GroundTruth] | [4 × 5 table] | [11,417 × 4 timetable] |
| AS 1600 | [1 × 1 GroundTruth] | [3 × 5 table] | [9568 × 3 timetable] |
| GA | [1 × 1 GroundTruth] | [8 × 5 table] | [1545 × 8 timetable] |
| Czajka MP02A | [1 × 1 GroundTruth] | [6 × 5 table] | [1843 × 6 timetable] |

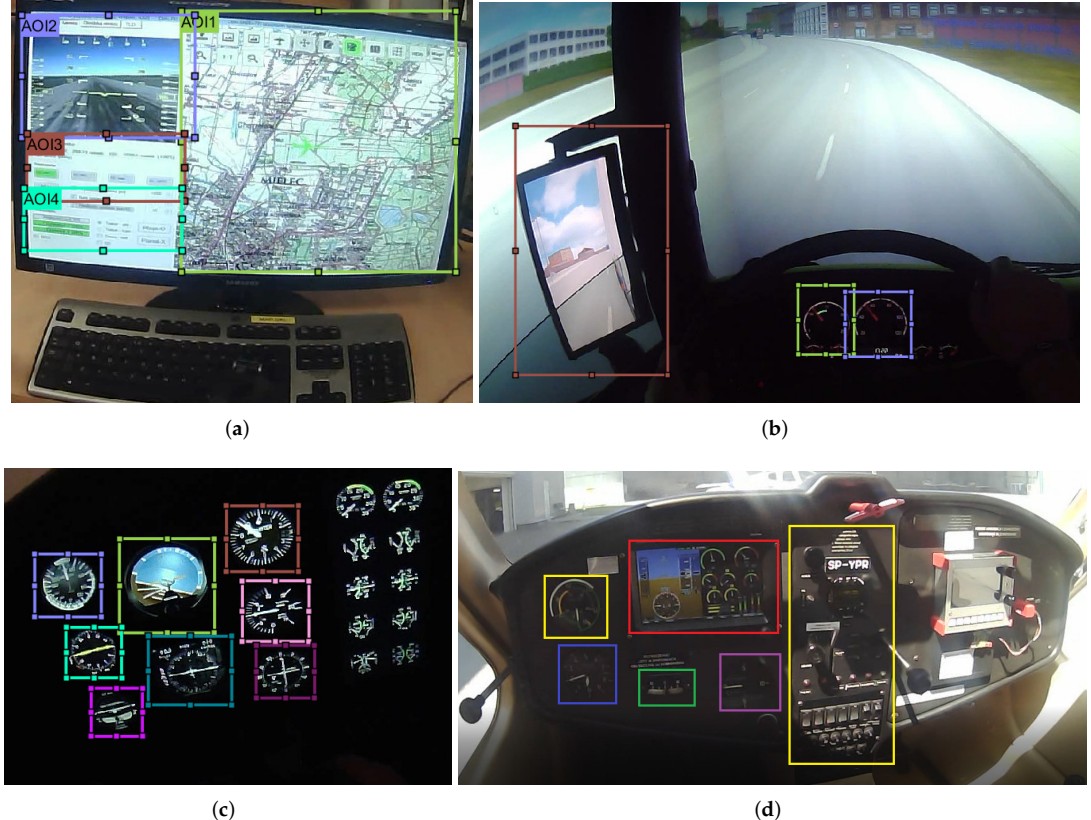

**Figure 5.** List of areas of interest that were used to build a training set for DNN networks, RPAS cockpit (**a**), truck simulator cockpit (**b**), GA cockpit (**c**) and flying lab cockpit (**d**).

As we already know, only a few layers in a deep neural network are responsible for selecting image features. In order to make easier observations of the input convolutional layer that extracts basic image features, such as the area or edge, it has been visualised in the form of a weight filter in Figure 6. Processing these features by deeper layers of

the net enables the extraction of further image features with a higher degree of cocpit instruments details. In order to emphasise the changes taking place during the training of the pre-trained RCNN network, a visualisation of the weight matrix of the convolutional layer has been included after completing the training process Figure 6a. Due to subtle differences in the details of the obtained weight distribution, a visualisation of the difference in the weight matrix before and at the end of the training process is presented in Figure 6b. The visualisation of the obtained difference demonstrates the influence of the training sequence used in the training process on the form of the weight matrix distribution in the convolutional layer. In this way, we can observe the influence of the applied training sequence in the learning process on the form of the input layer weight matrix and primitive features extraction process.

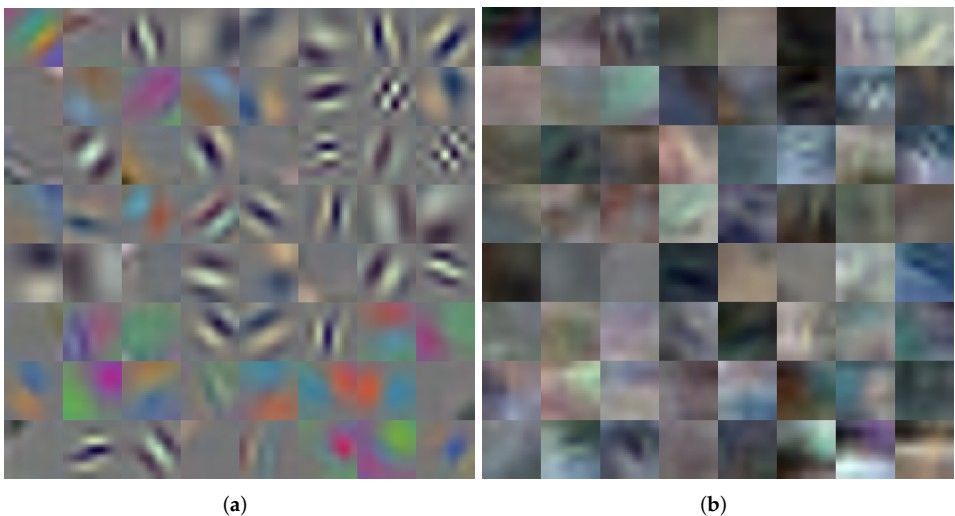

(**a**)　　　　　　　　　　　　　　　　　　(**b**)

**Figure 6.** The visualisation of the weight filter from the input convolutional layer in the R-CNN net after training (**a**) and difference with pretrained weight matrix (**b**).

## 4. Results and Discussion

Two object detection techniques using convolutional neural networks were selected: YOLO and R-CNN. Off-the-shelf GoogLeNet and SqueezeNet structures were implemented in each of them as modules responsible for detecting the position of instruments visible in the field of observation of the HMI system operator (see Table 2). They were then trained on prepared training data sets consisting of randomly selected and shuffled 500 frames (video stream, frame: $640 \times 480 \times 3$, 30 fps) from four different simulators, in which the position of the ROI was manually determined. In the experimental and measurement part, a series of tests have been carried out with the participation of 20 people who successively performed tasks defined by experts conducting training classes on individual simulators. For the GA and Czajka flight simulator, it was the task of landing on instruments (IFR) at the EPRZ Jasionka airport (the international ICAO code for civil airports annotations) in night conditions. For the RPAS simulator, it was a mission to perform an unmanned aerial vehicle flying in a defined geographical location around the airport in EPML Mielec. For the A1600 simulator, the operator's task was to perform a ride in a virtual city scenery, taking into account the road infrastructure and pedestrian traffic. For each recorded video stream, the appropriate video sequences were selected containing full imaging of the simulator instruments, the use of which by the operator will be further analysed. Using the modified KLT algorithm, which was described in the previous chapter, a training sequence was generated consisting of ROI coordinates for individual instruments tracked in the observed scene along the time axis of individual image frames for all selected video sequences. A total of 5000 training sets was created for each simulator, which were split on 70–20–10% parts, respectively, i.e., into training, validation and test sets in order to avoid the possible overfitting effect. Detection performance was tested on a video stream not included in the

training and validation sets. It was observed that for the adopted network models, the generation of region proposals is faster and better adapted to the data compared to other R-CNN models. In Table 3, the fourth and final stage of the learning process is presented.

**Table 2.** The network architectures considered in the experimental part of the study.

|  | R-CNN GoogLeNet | R-CNN SqueezeNet | YOLO GoogLeNet | YOLO SqueezeNet |
|---|---|---|---|---|
| Input type | Image | Image | Image | Image |
| Output type | Classification, Object detection | Classification, Object detection | Object detection | Object detection |
| Number of layers | 155 | 78 | 105 | 42 |
| Number of classifications | 183 | 87 | 122 | 45 |

**Table 3.** Re-training Fast R-CNN using updated RPN.

| Epoch | Iteration | Time Elapsed (hh:mm:ss) | Mini-Batch Loss | Mini-Batch Accuracy (%) | Mini-Batch RMSE | Base Learning Rate |
|---|---|---|---|---|---|---|
| 1 | 1 | 00:00:00 | 0.1412 | 96,88 | 0.10 | 0.0050 |
| 1 | 100 | 00:00:18 | 0.1430 | 100.00 | 0.14 | 0.0050 |
| 1 | 200 | 00:00:37 | 0.1068 | 98.44 | 0.14 | 0.0050 |
| 1 | 300 | 00:00:56 | 0.1377 | 93.75 | 0.08 | 0.0050 |
| 1 | 400 | 00:01:15 | 0.1731 | 95.31 | 0.09 | 0.0050 |
| 1 | 500 | 00:01:34 | 0.1245 | 100.00 | 0.11 | 0.0050 |
| 2 | 600 | 00:01:52 | 0.1997 | 93,75 | 0.11 | 0.0050 |
| 2 | 700 | 00:02:11 | 0.0565 | 100.00 | 0.08 | 0.0050 |
| 2 | 800 | 00:02:30 | 0.1225 | 96.88 | 0.10 | 0.0050 |
| 2 | 900 | 00:02:49 | 0.0942 | 98.44 | 0.10 | 0.0050 |
| 2 | 1000 | 00:03:07 | 0.0852 | 96.88 | 0.06 | 0.0050 |

For such a trained network, tests were carried out for individual simulators. The fixation coordinates indicated by the eyetracker were entered into the coordinate system of the individual video frames of the recorded stream. The neural network that conducted the detection of the presence of areas of interest in each frame of the video stream provided the position coordinates of the recognised instruments. During the learning process, it was found that for video frames containing a high concentration of instruments, the effect of incorrect identification of the type of instrument appeared. Therefore, as the assessment of the efficiency considered net architectures, we have obtained the $R_2$ coefficient values which have been ranging between 0.82 and 0.94. Since $R_2$ is highly sensitive to the presence of falsly identified instruments, we reported two more robust metrics for model performance evaluation: RMSE and MAPE. The R-CNN model showed lowest errors with RMSE ranged between 0.92 and 0.96 and MAPE ranged between 5.50% and 8.45%, respectively. The final result of the system is an observation histogram constructed by detecting the coincidence of fixation coordinates and coordinates of detected instruments, as shown in Figure 7. To construct the spatio-temporal trajectory, the mechanism of coincidence detection and contour blurring of the observed instruments described in [1,2] was used. Each frame from the recorded video stream can be used as the background of the resulting graph, regarding limitations that all instruments used by the operator involved in attention trajectory analysis are displayed correctly. The extracted contour lines show the individual ROIs defined in the analysed scene. The vertical axis of the chart shows the time of the observations, while the rising points represent successive, chronologically appearing fixations in the fields of individual instruments. The lines connecting the points are a virtual representation of the real saccades taking place in real time in successive frames of the recorded video stream. The space-time course of fixations is the basis for further expert analysis of the acquired efficiency of the system monitoring and control by the operator. From the point of view of the effectiveness of the training of operators of the systems involved in this work, it is important to carry out measurements under real operating conditions of the operator, taking into account the accompanying time deficit for making important operating decisions.

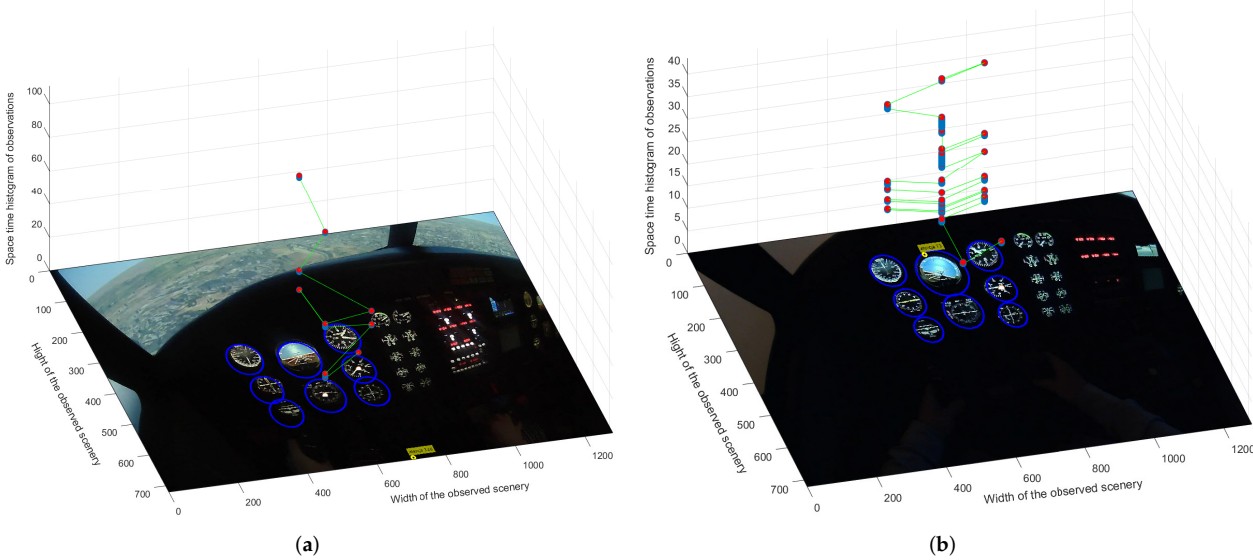

**Figure 7.** Spatio-temporal trajectory of the observer's attention fixation for a GA class simulator for VFR (**a**) and IFR conditions (**b**) respectively.

## 5. Conclusions

Using the video stream from the PupilLabs device and the DNN network architectures considered in this work, it is possible to construct a suitable training set to detect, in a video stream, the coordinates of instruments located in the operator's field of observation of selected HMI systems. A deep neural network allows for sustained instrument tracking in situations where classical algorithms stop their work due to introduced noise. To use R-CNN detectors in a real-time system, hybrid solutions should be sought, for example combining DNNs with tracking algorithms such as the Point Tracker KLT used in the paper. A neural network could initially or periodically recognise objects that appear in the field of view, while the Point Tracker could track their position with small movements of objects. For the YOLO SqueezeNet architecture, despite its high speed, no results were obtained to enable its effective use, as its efficiency was unsatisfactory. The Faster R-CNN architecture showed the highest identification efficiency. The disadvantage of this solution was the drastic increase in processing time for a single frame of the video stream. This fact significantly hinders its use in real-time object tracking solutions. The proposed system makes it possible to check the verification of practical skills at different stages of training. Using DNNs and the contour blurring mechanism, it is possible to detect in a video stream the locations of selected objects and, on this basis, effectively construct spatio-temporal fixation statistics. The key advantage of the developed method is the relatively fast and precise transfer of as much information as possible to the operator using systems where there is a need to control both machines in the broadest sense and processes. The three-dimensional time course of the observation coordinates, which is the actual trajectory of the operator's attention, provides the basis for assessing the level of training of the subject. This strategy universalises the proposed technology to different application areas and allows the ergonomics of different HMIs to be evaluated. However, the proposed method allows for an effective analysis of the observer's attention trajectory, so some of its limitations can be identified. The first one is the difficulty of implementing the proposed method in the real-time regime with the use of more complex deep neural network architectures due to the high computational complexity of the algorithm. Moreover, the question remains whether for the applied methods of network architectures and simulator sets it is possible to easily include other HMI platforms without reducing the achieved effectiveness of the currently trained network. Taking into account the above, it is necessary to conduct further experiments regarding other simulators using the HMI system. A natural development of the research

presented in this paper may be the use of the proposed technology to detect anomalies in the HMI system operator's work, for example their fatigue, improper chronology of activities checking the correct operation of the system and its control, etc. In particular, the proposed measurement method using DNN can be tested with the participation of flight controllers, during training on simulator stations. Currently, new directions of studies are concentrated on the use of non-invasive eyetrackers, which enable remote registration of the observer's attention in a way that does not require additional activity related to the need of use a specific type of glasses or micro camera sets for the operator.

**Author Contributions:** Conceptualisation, Z.G., E.Z., B.T., D.K. and P.R.; methodology, Z.G., E.Z., B.T., D.K. and P.R.; software, Z.G., E.Z., B.T., D.K. and P.R.; validation, Z.G., E.Z., B.T., D.K. and P.R.; formal analysis, Z.G., E.Z., B.T., D.K. and P.R.; investigation, Z.G., E.Z., B.T., D.K. and P.R.; resources, Z.G., E.Z., B.T., D.K. and P.R.; data curation, Z.G., E.Z., B.T., D.K. and P.R.; writing—original draft preparation, Z.G., E.Z., B.T., D.K. and P.R.; writing—review and editing, Z.G., E.Z., B.T., D.K. and P.R.; visualisation, Z.G., E.Z., B.T., D.K. and P.R.; supervision, Z.G.; project administration, Z.G., E.Z., B.T., D.K. and P.R. All authors have read and agreed to the published version of the manuscript.

**Funding:** This research received no external funding.

**Institutional Review Board Statement:** Not applicable.

**Informed Consent Statement:** Not applicable.

**Data Availability Statement:** Not applicable.

**Conflicts of Interest:** The authors declare no conflict of interest.

## Abbreviations

The following abbreviations are used in this manuscript:

| | |
|---|---|
| AGL | Above Ground Level |
| AMSL | Above Medium Sea Level |
| ANN | Artificial Neural Networks (ANN) |
| AOI | Area Of Interest |
| BVLOS | Beyond Visual Line of Sight |
| CNN | Convolutional Neural Betwork |
| DA | Decision Altitude |
| DH | Decision Height |
| DNN | Deep Neural Network |
| FFNN | Feedforward Neural Networks |
| GA | General Aviation |
| HMI | Human–Machine Interfaces |
| IFR | Instrument Flight Rules |
| ILS | Instrument Landing System |
| IoT | Internet of Things |
| KLT | Kanade–Lucas–Tomasi |
| ML | Machine Learning |
| RPAS | Remotely Piloted Aircraft System |
| QNH | Q Nautical Height |
| VFR | Visual Flight Rules |

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
