# Peer review of "Use of a DNN in Recording and Analysis of Operator Attention in Advanced HMI Systems"

_applsci, doi:10.3390/app122211431_

Round 1

Reviewer 1 Report

The authors have presented a smart technology to analyze the attention of operators of transportation devices using a deep neural network. They use a modified Kanade–Lucas–Tomasi algorithm for optimizing the labeling of the cockpit instruments of four simulators: General Aviation (GA), Remotely Piloted Aircraft System (RPAS), AS 1600, and Czajka. The scientific contribution to the deep neural network is quite OK to me. However, some literature reviews for recent deep learning should be discussed. The authors missed the point that the presented idea works well with automated machine learning (AutoML). For example, the following papers using AutoML should be mathematically added: AutoML to Date and Beyond: Challenges and Opportunities. ACM Comput. Surv. 54(8): 175:1-175:36 (2022), Human behavior in image-based Road Health Inspection Systems despite the emerging AutoML, Journal of Big Data 9 (1), 1-17, and Data Cleaning and AutoML: Would an Optimizer Choose to Clean? Datenbank-Spektrum 22(2): 121-130 (2022). Moreover, the visualization of the weight filter from the input convolutional layer in the R-CNN net is not so clear. It should be more elaborated clearly. The result of the method for an observation histogram constructed by estimating the coincidence of fixation coordinates and coordinates of detected instruments should be discussed in detail. In summary, I recommend that the paper should be accepted after suitable revision. If not, I am afraid to reject this paper.

Author Response

Thank you for your review of our work. Please find attached the responses to the comments submitted to the text of the manuscript.

Reviewer 2 Report

1.      Introduction should be more illustrative.

2.      Literature survey should be more comprehensive.

3.      A detailed discussion is required on the preprocessing of the data.

4.      Comparisons of the performance of the convolution neural networks should be analyzed using various performance metrics.

5.      Discuss the limitation of the study in a separate section.

6.      Future extension of the study should included in the conclusion section.

Author Response

(The authors gave the same response as above.)
